# Catch-A-Waveform: Learning to Generate Audio from a Single Short Example

**Gal Greshler**
Technion – Israel Institute of Technology
galgreshler@gmail.com

**Tamar Rott Shaham**
Technion – Israel Institute of Technology
stamarot@campus.technion.ac.il

**Tomer Michaeli**
Technion – Israel Institute of Technology
tomer.m@ee.technion.ac.il

## Abstract

Models for audio generation are typically trained on hours of recordings. Here, we illustrate that capturing the essence of an audio source is typically possible from as little as a few tens of seconds from a single training signal. Specifically, we present a GAN-based generative model that can be trained on one short audio signal from any domain (*e.g.* speech, music, etc.) and does not require pre-training or any other form of external supervision. Once trained, our model can generate random samples of arbitrary duration that maintain semantic similarity to the training waveform, yet exhibit new compositions of its audio primitives. This enables a long line of interesting applications, including generating new jazz improvisations or new a-cappella rap variants based on a single short example, producing coherent modifications to famous songs (*e.g.* adding a new verse to a Beatles song based solely on the original recording), filling-in of missing parts (inpainting), extending the bandwidth of a speech signal (super-resolution), and enhancing old recordings without access to any clean training example. We show that in most cases, no more than 20 seconds of training audio suffice for our model to achieve state-of-the-art results. This is despite its complete lack of prior knowledge about the nature of audio signals in general.

## 1 Introduction

In recent years, deep models for audio generation have had an immense impact on a wide range of applications, including text-to-speech synthesis [12, 38, 15, 7], voice-to-voice translation [8, 53], music generation [33, 11], singing voice conversion [8, 53], timbre transfer [16, 40], bandwidth-extension [29, 6], and audio inpainting [37]. Existing generative models require large datasets of training signals from the domain of interest. However, there are practical scenarios in which such datasets are extremely hard to collect, or are even nonexistent. Examples include a speaker that has only recorded a few sentences, an artist that had the chance to record only a few songs, or a unique jazz improvisation appearing in one particular recording. A natural question to ask, then, is whether large amounts of training data are a necessity for training a generative model.

Here, we take this question to the extreme. We illustrate that capturing the essence of an audio source is possible from as little as a few tens of seconds from a single training recording. Specifically, we present a generative adversarial network (GAN) based model that can be trained on one short raw waveform and does not require pre-training or any other type of external supervision [1]. Once the

---

[1]code is available at https://github.com/galgreshler/Catch-A-Waveform

35th Conference on Neural Information Processing Systems (NeurIPS 2021).

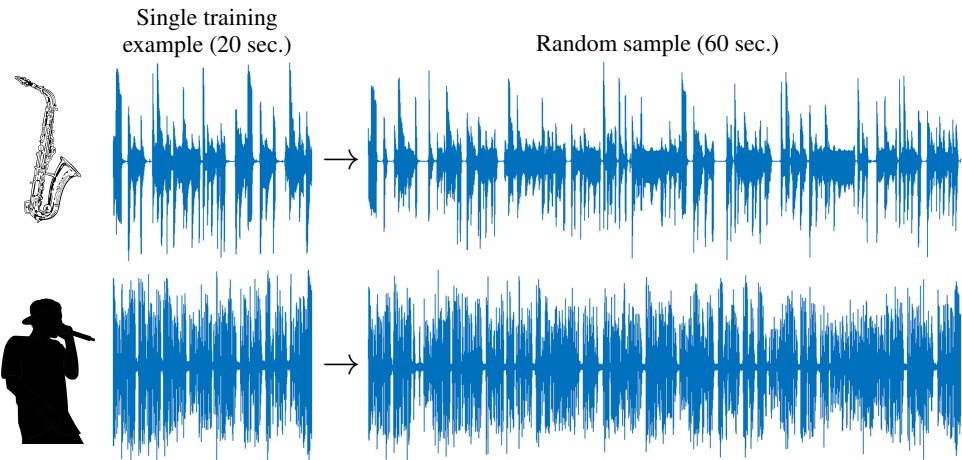

Figure 1: **Catch-A-Waveform.** We present a generative model that is able to capture the statistics of a single short audio recording (20 seconds in these examples). At inference, it can generate new diverse samples of arbitrary length, that exhibit new interesting compositions. The figure illustrates generation of new jazz improvisations and new freestyle rap variants. All examples can be listened to in our website.

model is trained, it is capable of generating diverse new signals that are semantically similar to the training recording, but contain new compositions and structures. Our model can handle different types of audio signals, from instrumental music to speech. For example, after training on 20 seconds of a saxophone solo, our model is able to generate new similar improvisations. The same can be done with a-capella rap, or old famous speeches, as exemplified in Fig.1. Our model can also generate samples conditioned on the low frequencies of some signal (be it the training signal or a similar one). This constraints the global structure of the generated signals, allowing to generate *e.g.* new versions of a Beatles song (all audio samples mentioned in the paper can be found in our website).

It is important to note that a short snippet of an audio signal is insufficient for learning language (for speech) or rules of harmony (for music). Therefore, our generated signals lack the linguistic semantics or long-range harmonic structure that can be potentially achieved with externally-trained models. However, surprisingly, the coherence of our generated signals over short time scales, typically suffices for confusing listeners to believe they are real, as we confirm through extensive user studies.

Besides generating random samples, we illustrate the utility of our approach in the common tasks of *bandwidth extension*, *inpainting* and *denoising* (see Fig. 2). We show that in the latter two tasks, no training signal whatsoever is required beyond the input itself. This allows handling sources for which no training data exist, like old recordings of famous musicians. In fact, our evaluation suggests that for the tasks of bandwidth extension and inpainting (sections 4.3 and 4.4), limiting the training to a single short signal is actually beneficial, and can lead to results that outperform models trained on hours of recordings.

Our work is inspired by generative models for visual data, which have been recently explored in the context of learning from a single image [50, 52] or a single short video [20]. Similarly to those works, we present a multi-scale GAN architecture that generates signals in their raw (time domain) representation. Audio signals, however, are very different from visual data; they are of high temporal resolution (usually at least 16,000 samples per second), they exhibit correlations at very long timescales, and they have diverse frequency contents. As we discuss, this necessitates dedicated architectures, losses, and adaptive selection of the multi-scale pyramid levels.

## 2 Related Work

**Generative models for audio.** Audio generation models have been extensively studied in the past few years. Some utilize autoregressive architectures [43, 39], including the computationally efficient inverse autoregressive flow (IAF) scheme [42, 47] and other flow based models [27, 48, 25, 49]. Others use GANs and variational autoencoders (VAEs), which have been found effective for many

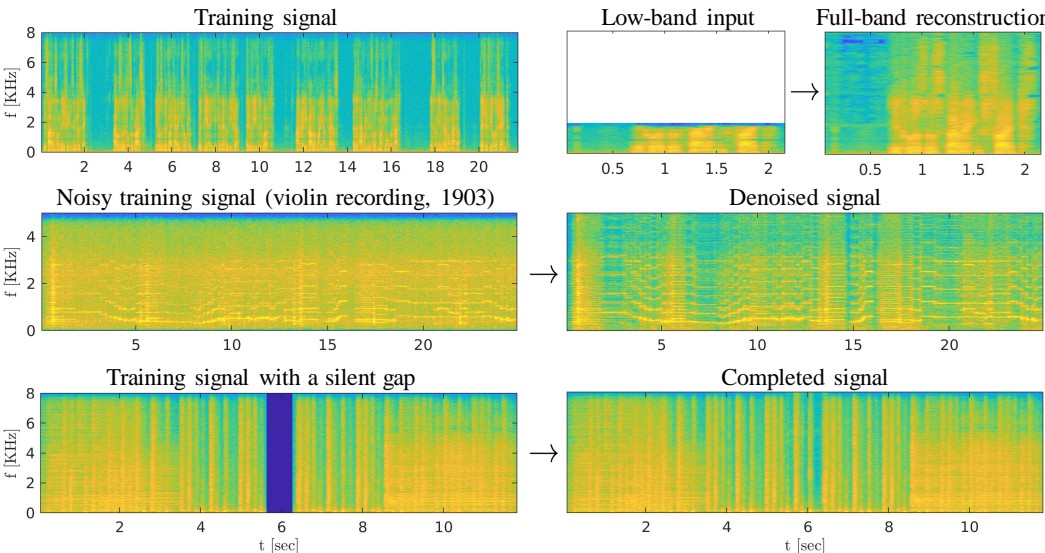

Figure 2: **Applications.** Our method can be used for a variety of tasks, including extending the bandwidth of a low-resolution signal, enhancing a noisy signal (without any prior knowledge on the signal or the noise), and completing missing parts.

applications, including text-to-speech [12, 38, 15, 7], unconditional generation [33, 11], singing voice conversion [8, 53], timbre transfer [40], inpainting [13, 37], bandwith-extension [26], and denoising [45, 32]. Several pipelines also integrate classical signal processing blocks to obtain improved results [16]. All these models rely on large training sets with hours of recordings. In contrast, here we focus on settings where only a single short signal is available for training.

**Few shot audio learning.** Audio generation models have also been taken to the few-shot regime, mainly in the context of voice cloning for speech [2, 9] and singing [41]. In this setting, only a few examples are provided at *test-time*. However, a *large training set* is still used for learning to perform the task. Here, on the other hand, we study the use of a single short waveform for training.

**Internal generative learning.** Exploiting the internal statistics of a single audio example by training a deep neural network (DNN) was recently explored for the tasks of audio restoration, source separation, audio editing, and ambient sound synthesis [62, 55]. These methods, however, cannot generate fake signals of complex structure (like music or speech). In the visual domain, recent *generative* models, like SinGAN [50] and InGAN [52], were developed for learning from a single natural image. These approaches were later extended to other domains, including videos [20], medical imaging [61], and 3D graphics [21]. Here we adapt some of these ideas to the audio domain.

## 3 Method

Consider a short sample $x$ from a stationary audio source. Our goal is to learn a generative model that can draw new random samples $\tilde{x}$ from the source's distribution. Our approach is inspired by the single image GAN (SinGAN) model [50]. Specifically, we aim at matching the distribution of length-$T$ segments of $\tilde{x}$ to that of length-$T$ segments of $x$, at multiple resolutions.

**Analysis pyramid.** We start by constructing an analysis pyramid of the training signal,

$$\begin{aligned} x_0 &= x, \\ x_n &= (x * h_n) \downarrow_{d_n}, \quad n = 1, \ldots, N, \end{aligned} \quad (1)$$

where $d_1 < d_2 < \cdots < d_N$ are down-sampling factors and $h_1, \ldots, h_n$ are the corresponding anti-aliasing filters. This is illustrated at the top of Fig. 3. Denoting the sampling rate of $x$ by $f^s$ (usually 16Khz in our experiments), we have that the sampling rate at the $n$th pyramid level is $f_n^s = f^s / d_n$. Similarly, we denote by $\tilde{x}_n$ the $n$th level of the multi-scale representation of the fake signal $\tilde{x}$.

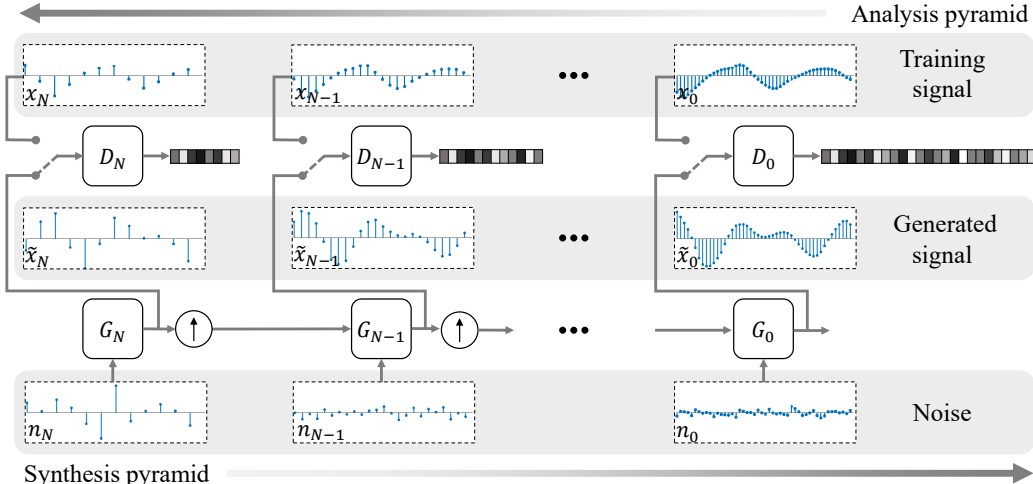

Figure 3: **Model illustration.** Our model is built from a pyramid of generators that operate at gradually increasing sampling rates, each fed by the preceding one. Adversarial training is performed sequentially in a coarse-to-fine manner, using a corresponding pyramid of discriminators.

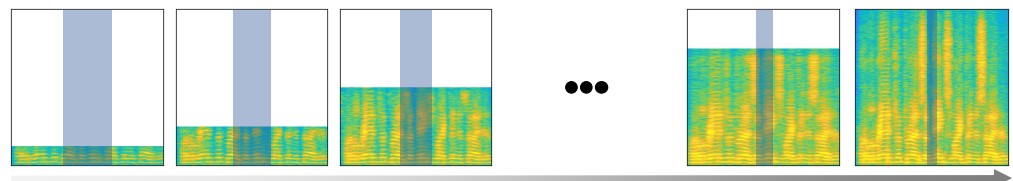

Figure 4: **Generation process.** Our generation process gradually increases the frequency range of the signal. The receptive field of all generators is the same. This translates to larger *effective* receptive fields (shaded rectangles) at the lower sampling rates, which shape the global structure of the signal.

**Synthesis pyramid.** The generation of a fake sample $\tilde{x}$ is performed sequentially by generating each of its pyramid levels conditioned on the previous one, from coarse to fine. Specifically,

$$\begin{aligned} \tilde{x}_N &= G_N(z_N), \\ \tilde{x}_n &= G_n\left(z_n, (\tilde{x}_{n+1})\uparrow^{\alpha_n}\right), \quad n = N-1, \ldots, 0, \end{aligned} \tag{2}$$

where $z_n$ is white Gaussian noise, $G_n$ is a convolutional neural network generator, $\alpha_n = d_{n+1}/d_n$ is the resolution ratio between scales $n+1$ and $n$, and $(\cdot)\uparrow^{\alpha}$ stands for up-sampling by a factor of $\alpha$ using cubic interpolation [24]. The signal $\tilde{x}_0$ at the end of this process is the generated fake sample $\tilde{x}$. This synthesis pyramid is shown at the middle and bottom rows of Fig. 3. All generators have the same receptive field, as measured in samples. This translates to larger effective receptive fields (in seconds) for the coarser levels than for the finer ones. As a result, the coarsest scale can capture the long-range dependencies that are typical of low frequencies of audio signals. Each subsequent generator, then, only needs to add a narrow band of frequencies to the signal generated at the previous scale (see Fig. 4). The higher the frequency band, the smaller the receptive field that suffices to achieve this goal. Following this understanding, we take the variance of $z_n$ to be proportional to the energy of $x$ in the frequency band $[\frac{1}{2}f_n^s, \frac{1}{2}f_{n-1}^s]$, which is at the responsibility of the generator $G_n$ to synthesize. It is important to note that as opposed to images, audio signals tend to exhibit long-range dependencies even at the highest frequency bands. Therefore, we take the receptive field in samples to be three orders of magnitude larger than the resolution factor $\alpha$ between scales (see below). This is as opposed to the SinGAN image model [50], which uses only one order of magnitude.

**Training.** We train our model in a coarse to fine manner as well. At each stage, a single generator in the pyramid is trained while the generators of all coarser levels are kept fixed. When training the $n$th level, the goal is to drive the distribution of length-$T$ segments within $\tilde{x}_n$ to become as close as possible to the distribution of length-$T$ segments within $x_n$. To this end, we use a patch-GAN

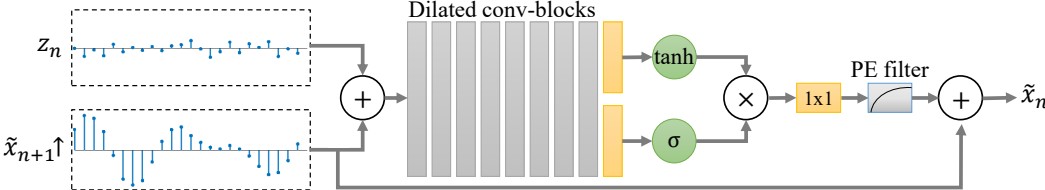

Figure 5: **Synthesis at a single scale.** The generator at the $n$th scale gets an up-sampled version of the signal generated at the previous scale, $(\tilde{x}_{n+1}) \uparrow^{\alpha}$, which has frequency contents in the range $[0, \frac{1}{2}f_{n+1}^s]$. Together with a noise realization $z_n$, it generates a signal $\tilde{x}_n$ with frequency contents in $[0, \frac{1}{2}f_n^s]$. This is done with a residual architecture involving 8 dilated convolution blocks; 7 of the form conv-BN-leakyReLU, 1 convolutional only. Dilation grows by a factor of 2 in each block. We add a gated activation at the end of the generator, followed by a fixed pre-emphasis filter.

framework [31, 22], which employs a convolutional discriminator network $D_n$ with receptive field $T$. The discriminator is tasked with classifying each of the overlapping length-$T$ windows in its input as real or fake, so that its output is a classification sequence of the same length as the input (minus $T - 1$ samples). The final score of the discriminator is the mean of this classification sequence. We specifically use the Wasserstein GAN loss [4],

$$\mathcal{L}_{\text{adv}}(D_n, G_n) = \mathop{\mathbb{E}}_{x \sim \mathbb{P}_{x_n}} [D_n(x_n)] - \mathop{\mathbb{E}}_{\tilde{x}_n \sim \mathbb{P}_{\tilde{x}_n}} [D_n(\tilde{x}_n)], \tag{3}$$

together with a gradient penalty [18]. Additionally, we pick a particular input at each scale, $z_n^r$, and enforce that its corresponding reconstructed signal, $\tilde{x}_n^r = G_n(z_n^r)$, be close to the real signal $x_n$ at that scale. This ensures that there is at least one point in the latent space of our model that maps to the real signal. We do this via a reconstruction loss,

$$\mathcal{L}_{\text{rec}}(G_n) = \alpha_1 \|x_n - \tilde{x}_n^r\|_2^2 + \alpha_2 \, \text{MSS}(x_n, \tilde{x}_n^r), \tag{4}$$

where the second term is the multi-scale spectrogram (MSS) loss [3, 42], which penalizes for differences between spectograms (thus disregarding phase). We use the particular MSS formulation of [11] (see SM). For the reconstruction sequences, we choose $\{z_N^r, z_{N-1}^r, ..., z_0^r\} = \{z^\star, 0, ..., 0\}$, where $z^\star$ is a fixed white Gaussian noise realization. Therefore, overall, we solve

$$\min_{G_n} \max_{D_n} \mathcal{L}_{\text{adv}}(D_n, G_n) + \mathcal{L}_{\text{rec}}(G_n), \tag{5}$$

where we alternate between performing one update step for $D_n$, which also involves minimizing the gradient penalty term, and one update step for $G_n$. In practice, we typically use only one of the terms in (4) (setting the other coefficient to 0), depending on the application (see Sec. 4).

**Architecture.** The generators and discriminators at all scales have the same fully-convolutional architecture. We use stacked blocks of 8 dilated convolutions, followed by batch normalization and leaky ReLU with slope $0.2$. The dilation factor grows by a factor of 2 in each layer, which is known to be an effective way for increasing of receptive field [60, 43]. At the end of the generator we use the gated activation unit [44], which is an element-wise product of tanh and sigmoid, each fed by an extra non-dilated convolution. All of our convolution layers have a kernel size of 9, which leads to a total receptive field of 2040 samples at each scale. We use weight normalization [51], which we found to improve results and training stability. At the end of the trainable blocks, we add a fixed pre-emphasis (PE) filter with impulse response $[1, -0.97]$, which amplifies the high frequencies, as common in similar tasks [58, 59]. An illustration of the generator's architecture is shown in Fig. 5.

**Automatic scales selection.** Different types of audio signals can have very different power spectra, as we illustrate in Fig. 6. This suggests that the frequency bands of the pyramid should be adaptively chosen based on the spectrum of the training signal. However, to allow for efficient implementations of resampling techniques, we also want the sampling rates of all scales to be rational factors (with small denominators) of $f^s$ [14, ch. 9]. We therefore use a predefined discrete set of potential sampling rates, and choose our bands adaptively only from this set. As can be seen in Fig. 6, up to 2Khz, where most of the audible energy resides, the predefined scales grow at a factor of around $1.25$. The mid-range, 2-4Khz, typically contains less energy and so the scales are sparser there. Finally, to be able to capture the energy of non-vocal syllables in speech signals, the scales become denser again

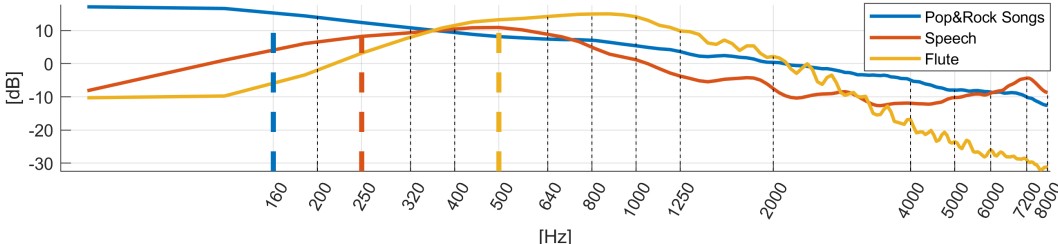

Figure 6: **Scales selection.** The plot depicts the power spectral densities of three different audio datasets (rock and pop songs [56], speech [23] and monophonic flute [63]). The dashed black lines show the predefined band partitions (note the logarithmic axis). The first band is adaptively chosen to contain enough energy. The bold colored lines show the typical first scale chosen for each dataset.

from 4Khz. In practice, the most significant effect is due to the automatic selection of the first band, which shapes the global structure of the signal. We therefore choose automatically only this band, such that it contains enough energy (see SM). Fig. 6 shows typical selections of the first band for different types of audio signals. Additional spectra are presented in the SM.

## 4 Experiments

We test our *catch-a-waveform (CAW)* method in several applications and evaluate it both qualitatively and quantitatively. Our training examples contain a variety of audio types, including polyphonic rock and pop music, monophonic instrumental music, speech, and ambient sounds. Unless noted otherwise, all training signals have a sampling rate of 16Khz. For training, we use the Adam optimizer [28] with $(\beta_1, \beta_2) = (0.5, 0.999)$ and learning rate 0.0015, which we reduce by a factor of 10 after two thirds of the epochs (we run a total of 3000 epochs). Training on a 25 second long signal takes about 10 hours on Nvidia GeForce RTX 2080. Inference is 60 times faster than real-time.

### 4.1 Unconditional generation

**Monophonic music.** We trained CAW models on monophonic music played by various instruments, including cello, violin, saxophone, trumpet, and electric guitar. Here, we used $\alpha_1 = 0$ and $\alpha_2 = 10^{-4}$ in (4). We trained on signals of length 25 to 100 seconds, and at test time generated signals of various lengths by simply injecting input noise signals of appropriate length (see SM for additional details). The generated signals sound like variations or naive improvisations on the original piece (see website).

**Speech signals.** We further trained CAW models on various human voice recordings, with lengths varying from 20 to 40 seconds. These include short segments from speeches of American presidents Trump and Obama and a-capella rap. Here we used $\alpha_1 = 10, \alpha_2 = 0$ in (4). At inference, we generated random samples of lengths between 20 and 60 seconds. As exemplified in our website, the generated signals preserve the voice of the speaker, but exhibit new compositions of syllables, words, intonations and silent gaps. Note that since our model has no notion of language, the generated signals are not necessarily interpretable. The temporal coherence of the generated signals can be controlled by changing the receptive field of the model. As we illustrate in the website, reducing the receptive field from 4 to 2 seconds (by removing one convolutional block), causes the structure to become less coherent and makes the generated speech sound like mumbling. Increasing the receptive field to 8 seconds, on the other hand, preserves short sequences of words from the training signal.

**Human perception tests.** In order to evaluate the perceptual quality of our generated signals, we conducted auditory studies through Amazon Mechanical Turk (AMT). The studies were performed on solo signals of 8 different instruments (saxophone, trumpet, violin, flute, clarinet, cello, accordion and distorted electric guitar) randomly chosen from the Medly-solos-Db dataset [34] and the solo-audio dataset [63]. For each instrument, we randomly cropped 25 second long segments from 7 different parts of the recording to serve as our real signals (56 segments in total) and used our method to generate a 10 second long fake version for each of them. We performed two types of user studies: (i) *a paired test*, where the real signal and its fake version were played sequentially, and the user was asked to choose which is the fake, and (ii) *an unpaired test*, where the user listened to a single signal and had to determine whether it is real or fake.

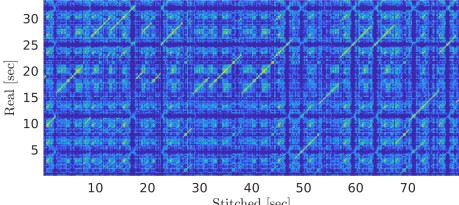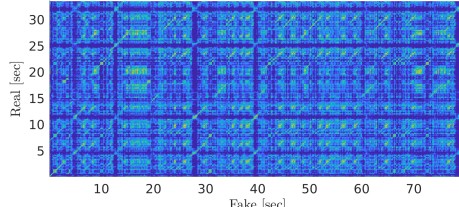

Figure 8: **Similarity matrix.** Distinct diagonal lines correspond to segments in the generated signal that were 'copied' from the real signal. Naively stitched signals show clear lines in the copied parts (left). In our generated signal (right) more lines can be seen as the model can maintain global temporal structure while combining higher frequencies from different temporal locations.

Each test opened with a tutorial of 5-8 questions identical to the structure of the main test, but with a feedback to the user. We also had an additional version, (iii) *an unpaired test with a paired tutorial*. In this case users were exposed to examples of paired real and fake signals during the training phase, but the test itself was unpaired. In each of the tests, we had $50$ different users answer $25$ questions each. The results are summarized in Fig. 7. As can be seen, in all the studies the confusion rates are relatively high (the ultimate rate being $50\%$). As expected, the confusion rate of the paired test is lower than the unpaired test, as this setting is less challenging. But there is no significant difference between the results of the two unpaired tests, suggesting that a paired tutorial does not help the listener perform better discrimination.

**Comparison to naive copy-and-paste.** As our model is trained on a very short signal, its ability to generate new semantic contents is limited. For example, it will most likely not generate syllables or notes that did not exist in the training signal. However, does that mean it naively copy and pastes segments from the training signal? To examine this, we depict in Fig. 8 (right pane) a similarity matrix between small overlapping patches of a generated signal and the training signal. We specifically use the cosine similarity between the vectors of absolute values of the discrete Fourier transforms of the patches (see more details about the similarity matrix calculation in the SM). In this visualization, matching segments appear as diagonal lines. For comparison, we also show the same visualization for a naively generated signal constructed by stitching together cropped segments from the training signal, with crop sizes randomly chosen between the receptive fields of our finest and coarsest scales. Stitching is done using cross fading. We ensured that the naively stitched signals sound realistic by performing another unpaired user study, as in the left bar of Fig. 7, and got a confusion rate of $47.88\%$. While the similarity matrix of the naively stitched signal shows clear solid lines, in our generated signal we can observe sometimes two or more parallel, weaker, lines one above the other. This suggests that our model often mixes several parts from the training signal. This happens thanks to our multi-scale architecture that allows each generated frequency band to contain contents from a different temporal location in the training signal.

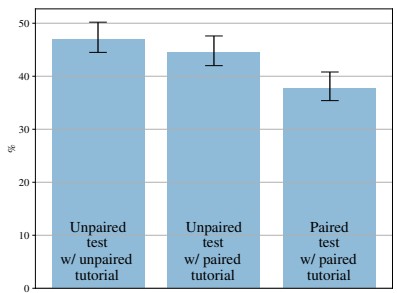

Figure 7: **Real-vs.-Fake AMT studies.** Users were asked to discriminate between our generated signals and real ones, both in paired and in unpaired tests. We present confusion rates and $[5\%, 95\%]$ confidence intervals. As can be seen, the confusion rates in all cases are close to the ultimate rate, which is $50\%$.

## 4.2 Conditional generation of music variations

Another interesting application is generating variations or extensions of existing songs (*e.g.* adding a new verse). To do so, we first train our model on a popular rock or pop song. Then, at inference time, we start the generation from the second coarsest scale by injecting the real (training) signal as input to that scale. This ensures that the generated signal maintains the global structure of the real signal, as its low frequencies are constrained to be the same. But finer details, like the lyrics, are randomly generated. Here we take $\alpha_1 = 0$ and $\alpha_2 = 10^{-4}$ in (4). Also, to encourage large variability between different random samples, we set the input noise in the second coarsest scale to have the same energy

| Original verse (used for training) | Generated verse |
|---|---|
| It's been a hard day's night | It's been a hard day's **day** |
| and I've been working like a dog | **chi get ya money to ba log** |
| It's been a hard day's night | It's been a hard **all day** |
| I should be sleeping like a log | **you got me working** like a **dog** |

Figure 9: **Music variations.** After training on a specific song, we can inject a down-sampled version of the song to the second coarsest scale of the model. This way, our model generates a signal having the same structure as the original song, but with randomly generated finer details, like lyrics. In this example we generate a new verse to "A Hard Day's Night" by The Beatles, after training on its two first verses. Modified lyrics are shown in red.

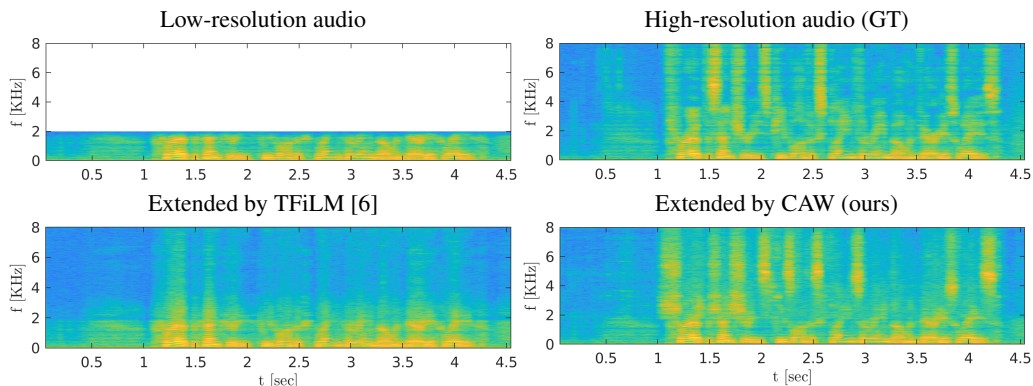

Figure 10: **Bandwidth extension.** We use our model for speech bandwidth-extension. A model trained on one short high-bandwidth signal (25 seconds in this case) can be used at test time to increase the bandwidth of any low-bandwidth signal of the same speaker (by injecting it to a coarse scale of the model). This results in sharper reconstructions than those obtained with TFiLM, which was trained on hours of examples.

as that of the real signal at that scale. Examples can be found in the website and in Fig. 9, which shows a new verse generated by our model to the famous Beatles song "A Hard Day's Night".

### 4.3 Bandwidth extension

Bandwidth Extension (BE) is the task of reconstructing a high-bandwidth signal from its low-bandwidth version, and is usually demonstrated on speech [29, 6, 26, 19, 57] and music [30, 54]. To perform BE using CAW, we first train it on a high-bandwidth short audio example of a specific speaker. At inference time, we can then inject any other low-bandwidth signal of the same speaker to a coarse scale of the model (we choose the scale whose sampling rate matches that of the input signal). Here we use $\alpha_1 = 10$ and $\alpha_2 = 0$ in (4). We then stitch the reconstructed higher frequencies generated by our model with the low frequency range of the input signal to obtain our final full-bandwidth reconstruction. Figure 10 shows a BE example, where the sampling rate of a speech signal is increased from $4$KHz to $16$KHz. Our bandwidth-extended signals contain realistic high frequency contents, which makes them sound sharp (see examples and comparisons in our website, including for the easier task of extension from $8$KHz to $16$KHz).

We compare our BE results to the the state-of-the-art temporal FiLM (TFiLM) method [6], which requires a large training set to perform this task. We use the VCTK dataset, and report both the signal to noise ratio (SNRs) and the log spectral distance (LSD) [17] between the recovered signal and the ground-truth one, averaged over a test set. LSD is known to better correlate with human perception. We perform comparisons to several TFiLM variants, following the protocols of [6].

Table 1: **Bandwidth extension quantitative evaluation.** We compare our method to TFiLM [6] using SNR (higher is better) and LSD (lower is better) both for multi-speaker test and single-speaker test. In all cases our model achieves better LSD scores, indicating of higher perceptual quality.

| | Multi speaker test | | | | Single speaker test | |
| | TFiLM [6] | | | CAW (ours) | TFiLM [6] | CAW (ours) |
| Training set size [min] | 25 | 240 | 600 | 0.4 | 30 | 0.4 |
| --- | --- | --- | --- | --- | --- | --- |
| SNR [dB] $\uparrow$ | 14.66 | 14.83 | 15.45 | $13.8 \pm 0.94$ | 14.77 | $13.03 \pm 0.83$ |
| LSD $\downarrow$ | 4.96 | 3.89 | 3.79 | $2.97 \pm 0.26$ | 3.92 | $3.03 \pm 0.26$ |

**Single-speaker baseline.** In this setting, we train a separate CAW model for each of 9 speakers, and then test each of the models on a set of held-out sentences of the same speaker. For TFiLM, we use 30 minutes of training data for each speaker, and for our model we use only 25 seconds. As can be seen in Table 1, our model outperforms TFiLM in LSD, but achieves a slightly lower SNR (we report mean and standard deviation over 50 different trained models).

**Multi-speaker baselines.** Here, we train TFiLM on 99 speakers from the VTCK dataset and test it on the remaining 9 speakers. We have three variants, corresponding to training sets of 25 minutes, 4 hours, and 10 hours. Our model is trained as in the single-speaker case. We use the same test set for evaluating both methods. As can be seen in Table 1, our model is again superior in terms of LSD compared to all TFiLM variants, and is slightly worse in terms SNR.

## 4.4 Audio inpainting

Audio inpainting refers to the task of completing a missing part of a given audio signal. It has been previously addressed using classical signal processing methods [1, 5, 35], graph-based approaches [46] and neural networks [13, 36, 37]. Here, we address the long-inpainting task, where several hundred milliseconds are missing. We do this by training CAW with slight adaptations: (i) we calculate the loss with respect to only the valid parts of the signal (excluding the gap), and (ii) we sample a new reconstruction noise realization for the missing part at each iteration. Here we use $\alpha_1 = 10$ and $\alpha_2 = 0$ in (4). After training, we take the completed part

Table 2: **Inpainting AMT study.** Users chose between our model (trained on 12 seconds), GACELA (trained on 8 hours), and the ground-truth signals. The preference rates indicate that our results are at least comparable to GACELA, and are often confused to be real signals.

| Study | Preference rate |
| --- | --- |
| Ours vs. GACELA | $55.3\% \pm 2.4\%$ |
| Ours vs. Real | $44.3\% \pm 2.3\%$ |

from the reconstruction, and stitch it with the input. As can be seen in Fig. 11, our model coherently completes the missing part, and thanks to its relatively large receptive field, the completion smoothly fuses with the valid parts. Examples of completed rock songs can be found in our website.

**Human perception tests.** We evaluated our results using an AMT user preference test. We took 64 rock songs from the FMA-small dataset [10], extracted a 12 second long segment from each, and masked a 750ms window. We compared our results with those of GACELA [37], a GAN based context encoder trained on roughly 8 hours of rock songs from the same dataset. In each query, raters listened to a 5-9 second long segment containing the missing gap, as well as to our and to GACELA's completions. They could re-listen to all signals as many times as they wanted, and eventually had to pick the completion that sounded better. In total, 50 raters answered 20 queries each. As can be seen in Table 2, raters preferred our completions over 50% of the times, suggesting that the performance of our method is at least comparable to GACELA's. We also performed a user study that compared our completions to the real signals. Interestingly, the preference rate for our completion was still relatively high (see Table 2), indicating that our results are often indistinguishable from real signals.

## 4.5 Audio denoising

An interesting side effect of CAW's training process, is that it can be used for audio denoising. As explained in Sec. 3, during training we enforce a certain noise hypothesis to generate a reconstruction of the training signal. We found that when training our model on a noisy signal, this reconstruction often preserves harmonic parts while suppressing ambient noise. This effect is more distinct when

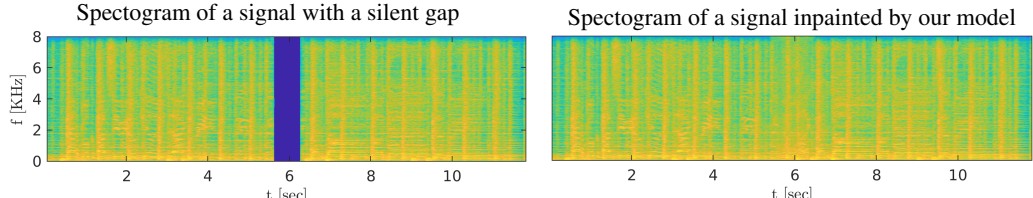

Figure 11: **Audio inpainting.** Our model is able to complete a mising silent gap in a given signal without any additional information other than the signal with the gap itself. We train our model on the valid region of the signal excluding the missing part, to learn its internal statistics, and then at test time we generate the missing gap, which results in a coherent realistic completion.

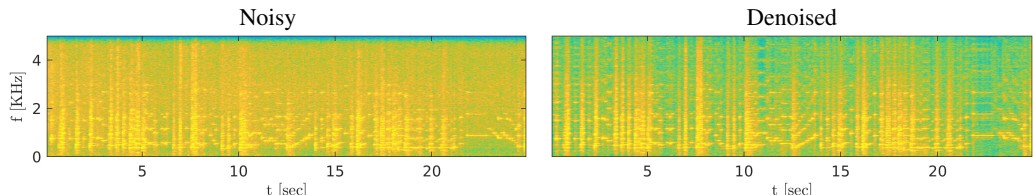

Figure 12: **Denoising.** When trained on a single noisy example, our model produces a clean reconstruction. This enables to denoise *e.g.* old recordings, illustrated here on one of the first violin recordings, from 1903. Here, no ground-truth data is available for computing SNR. However, in controlled experiments with modern recordings of the same musical piece, we measure improvement of $1.5\text{dB} - 4\text{dB}$ in SNR (see text for details).

using $\alpha_1 = 10$ and $\alpha_2 = 0$ in (4). This enables to perform denoising without access to a any clean example for training, and without any prior knowledge about the noise distribution. This is in contrast to externally supervised approaches, which require many pairs of noisy-clean examples, *e.g.* [32]. As an example, we demonstrate denoising of old recordings of the violinist Joseph Joachim from 1903 (for which obviously no clean training examples can be collected). As seen in Fig. 12 and on our website, the reconstructed signals are notably cleaner than the original ones. To obtain some quantitative measure of the denoising performance in this experiment, we took a (clean) modern recording of the same musical piece (Bach's Adagio), preformed by famous violinist Hilary Hahn[2]. We synthetically generated noisy versions of this recording using both white noise and old gramophone noise recordings[3] (see the SM and our website). For each noise type, we examined noise levels of 5dB and 10dB. For the white noise, our denoising increased the SNR from $5\text{dB} \rightarrow 9.78\text{dB}$ and from $10\text{dB} \rightarrow 11.53\text{dB}$. For the gramophone noise, it increased SNR from $5\text{dB} \rightarrow 6.89\text{dB}$ and from $10\text{dB} \rightarrow 11.56\text{dB}$. We believe these results can be further improved in the future by optimizing the model for the specific task of denoising.

## 5  Conclusion and limitations

We presented a new GAN-based model for audio generation that can be trained on a single short example. Our model works on raw waveforms, and is useful for a variety of tasks. As we illustrated, learning from a single waveform often has advantages over learning from large datasets. We believe this new learning scheme can be further developed to a general framework for training 'personalized' deep learning models, where personal small data would be sufficient for a variety of tasks. This approach, however, is not free of limitations. Models trained on small data cannot learn high level semantics. Moreover, they can sometimes struggle even with low-level attributes. For example, our model is occasionally less successful in handling high-pitched speech signals, like that of a female or a child. Another challenging scenario is speech recorded in reverbrant environments (*e.g.* in a large conference room), where our model tends to transform the reverberations into high-pitched noise. Examples for a variety of such failure cases can be found in our website.

---

[2]taken from https://www.youtube.com/watch?v=c3mwVaQIZ1c
[3]taken from https://freesound.org/people/lollosound/sounds/387005/

**Acknowledgements**  This research was supported by the Israel Science Foundation (grant 852/17) and by the Technion Ollendorff Minerva Center.

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
