# Catch-A-Waveform: Learning to Generate Audio from a Single Short Example
# Supplementary Material

**Gal Greshler**
Technion – Israel Institute of Technology
galgreshler@gmail.com

**Tamar Rott Shaham**
Technion – Israel Institute of Technology
stamarot@campus.technion.ac.il

**Tomer Michaeli**
Technion – Israel Institute of Technology
tomer.m@ee.technion.ac.il

Code is available here. Audio samples and additional figures can be found on the project's website.

## 1 Model and training details

### 1.1 Training details

**Gradient penalty.** In each update step of the discriminator, we minimize the generator's loss with an additional gradient penalty regularization term [2], defined as

$$\lambda \mathop{\mathbb{E}}_{\hat{x} \sim \mathbb{P}_{\hat{x}}} [(\|\nabla_{\hat{x}} D_n(\hat{x})\|_2 - 1)^2], \tag{1}$$

where $\lambda = 0.01$ and $\hat{x}$ is a convex combination of the real signal $x_n$ and a generated one $\tilde{x}_n$, with random weights.

**MSS loss.** The MSS reconstruction loss we use, is given by

$$\text{MSS}(x_n, \tilde{x}_n^{\text{r}}) = \frac{1}{M} \sum_{m=0}^{M-1} \|\,|\text{STFT}_m(x_n)| - |\text{STFT}_m(\tilde{x}_n^{\text{r}})|\,\|_2 \tag{2}$$

where STFT is the short-time Fourier transform and $M$ is the number of different STFT parameter sets. The set of parameters we use are as follows:

| window size | hop length | fft size |
|---|---|---|
| 240 | 50 | 512 |
| 600 | 120 | 1024 |
| 1200 | 240 | 2048 |

### 1.2 Scales selection

As explained in the main text, we have a set of predefined sampling rates. The first scale of the model is chosen automatically among them, according to the signal energy at that scale. Specifically, we normalize the input signal $x$ such that $\max_n |x[n]| = 1$. Then, we choose the coarsest scale (namely scale $N$) to be the first that satisfies

$$\frac{1}{K} \sum_{n=0}^{K-1} x_N^2[n] \geq 0.0025, \tag{3}$$

35th Conference on Neural Information Processing Systems (NeurIPS 2021).

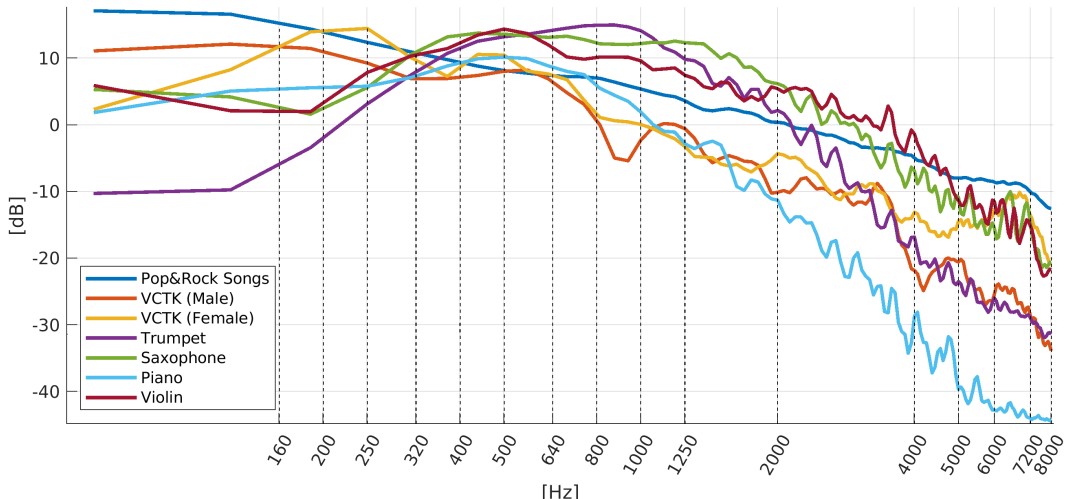

Figure 1: **Frequency contents of different datasets.** Note that complex music (here rock and pop songs) have a more energy in the low frequencies than speech and monophonic music.

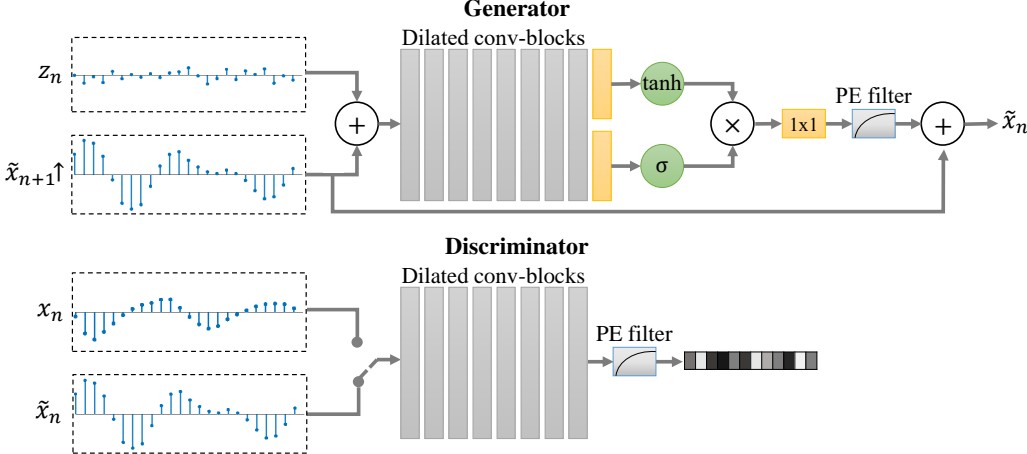

Figure 2: **Architectures.**

where $x_N$ is the real signal at that scale and $K$ is the number of samples in $x_N$.

Figure 1 shows the average frequency contents of several datasets, and the predefined scales. Note that the graph shows averages on entire datasets, while the actual first scale is chosen for each signal individually. The coarsest scale defines the receptive field (in seconds) for the entire model. Therefore, for inpainting tasks, we also make sure that the signal at the coarsest scale has more samples than the missing gap plus the receptive field.

## 1.3  Architecture

The generator at each scale is built from 8 dilated convolutional blocks. The first 7 blocks contain dilated convolution, Batch-Norm and leakyReLU with slope 0.2. The last block is convolutional only. The dilation factor grows exponentially from 1 in the first block to 128 in the last one. The convolutional blocks feed a gated activation unit, which is followed by an extra $1 \times 1$ convolution and PE filter. The discriminator at each scale is similar to the generator, except it does not have the gated activation unit. The number of channels in each layer, for both the generator and the discriminator, is 16 at the coarsest scale and 96 at the rest of the scales. Figure 2 shows an illustration of the generator and discriminator architectures.

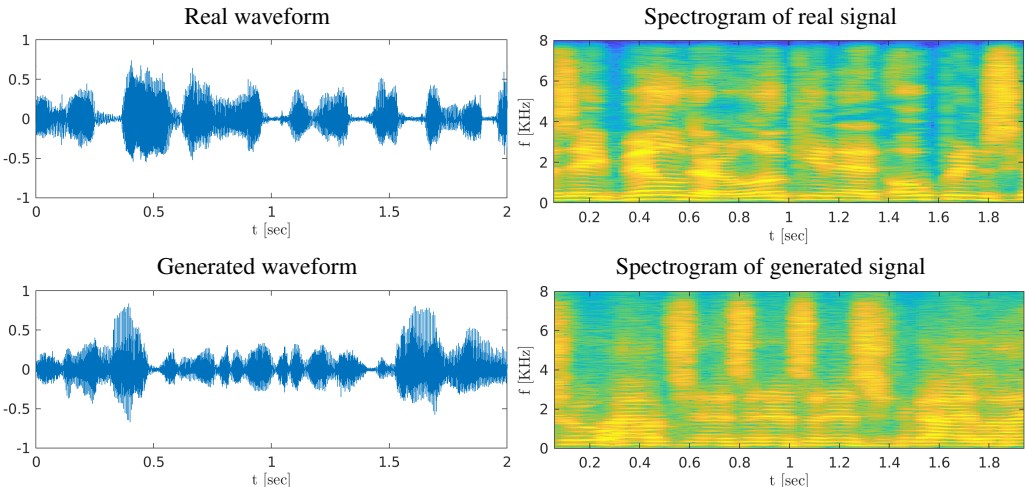

Figure 3: **Unconditional generation of speech signal**

## 1.4 Positional encoding

In order to ensure that the generator's output is of the same length as the real signal during training, we use zero-padding at its input. This zero padding functions as a positional encoding [4], which allows the generator to know the absolute location with respect to the beginning and ending of the signal. This encoding extends up to one receptive field from the beginning and one receptive field from the ending of the signal. Therefore, the generator manages to "remember" these parts and to "paste" them at the borders of the generated signals. This makes the beginning and ending of the generated signals sound like those of the input. If desired, this phenomenon can be avoided by simply trimming the generated signal by one receptive field from each side.

## 2 Additional experimental details

### 2.1 Unconditional generation

During training, we generate fake signals having the same length as the input. We do this by injecting to the generator noise of the same length as the input, padded with zeros of the length of the receptive field (we use no padding within the convolutional layers). To generate a signal of different length at inference time, we simply inject noise having the desired length at the coarsest scale's input. Figures 3-5 show examples of real and generated signals of different types. In order to evaluate performance and perceptual quality of our generated signals, a user study was conducted. Screenshots from the unpaired user study can be found in Fig. 9, and from the paired one in Fig. 10.

**Calculation of similarity matrix.** As explained in the main text, in order to better understand the nature of our generated signals and specifically how they differ from signals generated by a naive cut-and-paste approach, we compute a similarity matrix between the fake and the real signals. The matrix is calculated as follows. First we compute STFT matrices for the real and fakes signals, and take the absolute values of their entries. We denote these by $R$ and $F$, respectively. The STFT matrix is calculated on segments of 4096 samples, multiplied by the Hann window, and with hop size of 128 samples. Next, the similarity value between frame $i$ in the real signal and frame $j$ in the fake signal is computed as the cosine similarity between the $i^{th}$ and $j^{th}$ columns in $R$ and $F$, respectively, i.e.,

$$\text{sim}(i, j) = \frac{\langle R_i, F_j \rangle}{\|R_i\| \|F_j\|} \tag{4}$$

In Figure 6 we show several examples of similarity matrices of naively stitched signals and of our generated signals.

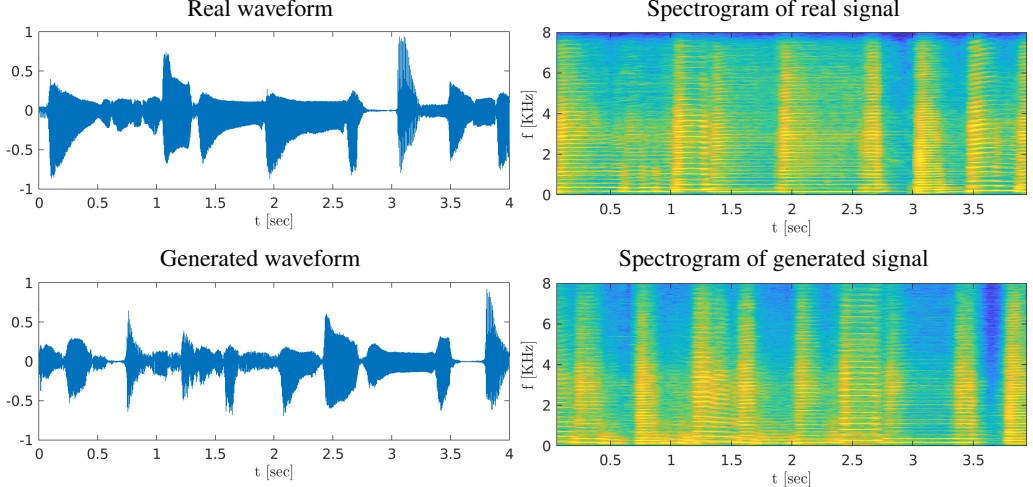

Figure 4: **Unconditional generation of saxophone signal**

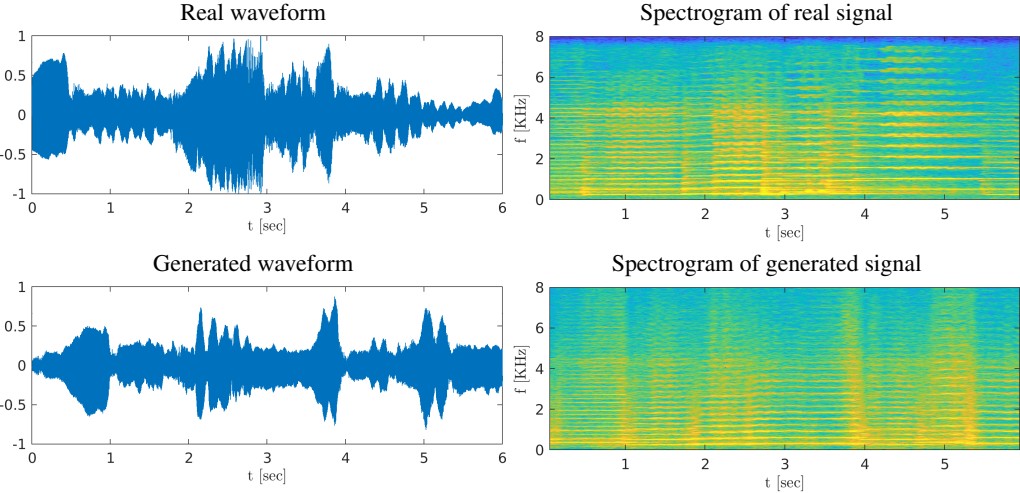

Figure 5: **Unconditional generation of violin signal**

## 2.2 Bandwidth extension

For the bandwidth extension experiments, we trained CAW models for each speaker of the 9 test speakers in the VCTK dataset. Each speaker's sentences were divided to batches of 4-10 sentences, such that each batch contains between 20 and 25 seconds of speech. This resulted in around 50 models for each speaker. At inference time, each sentence of the speaker was extended by all of the models, except for the one the sentence was trained on. For each sentence we calculated the mean result and the standard deviation, across all models. The mean and std reported in the main text correspond to the average of all sentences of all speakers. In the **single speaker** task, we only evaluate the sentences defined as test set for the TFiLM model [1].

**Evaluation metrics.** We used two common evaluation metrics in order to evaluate the BE results: SNR and LSD. These are defined as

$$\text{SNR}(x, \hat{x}) = 20 \log_{10} \left( \frac{||x||_2}{||x - \hat{x}||_2} \right)$$

$$\text{LSD}(x, \hat{x}) = \frac{1}{L} \sum_{l=1}^{L} \sqrt{\frac{1}{K} \sum_{k=1}^{K} \left( X(l, k) - \hat{X}(l, k) \right)^2}$$

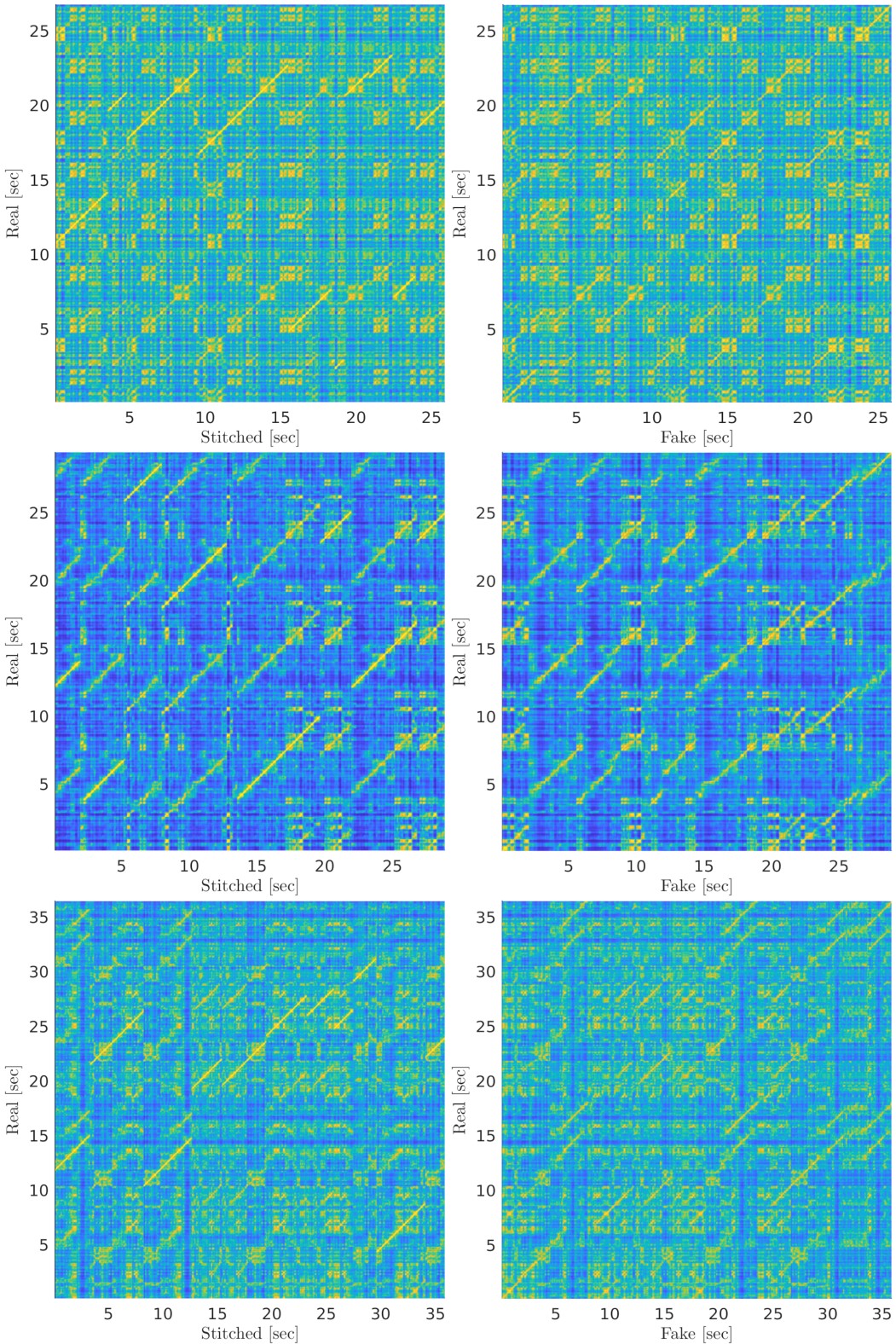

Figure 6: **Similarity matrices.** Matrices of signals created by naive cut and paste method (left column), and of our generated signals (right column). Our signals show more blurry lines as they can contain information from different temporal positions across frequency scales.

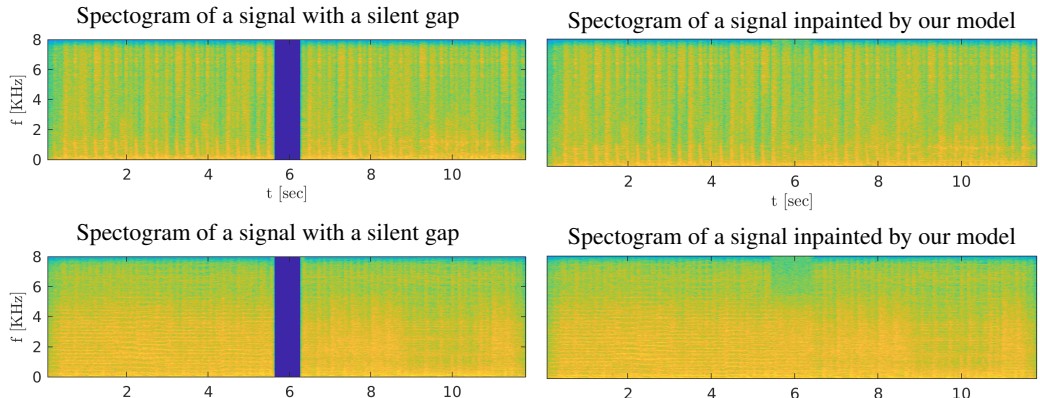

Figure 7: **Audio inpainting.** Examples of inpainting done by our model. The only input to the model is the signal with the missing gap, and a mask indicating the temporal location of the hole.

where $X$ and $\hat{X}$ are the log magnitudes of the STFTs of the ground truth signal $x$ and the output extended signal $\hat{x}$, respectively. $L$ is the number of STFT frames and $K$ is the window size, which is 2048 samples in our case, calculated without overlaps.

### 2.3 Audio inpainting

As explained in the main text, inpainting is done by training on a signal with a silent gap, where the loss terms are calculated only on the valid parts. More examples for inpainting of rock songs from the FMA dataset can be found in Fig. 7. In order to evaluate inpainting performance, a user study was conducted, comparing our results to the GACELA model [3] and to ground truth signals. Screenshots from the study can be found in Fig. 11

### 2.4 Audio denoising

As detailed in the main text, we examined denoising of noisy signals that we created by adding white noise and recorded gramophone noise to a clean violin recording. The clean signal, recorded gramophone noise and results of the method are presented in Figure 8.

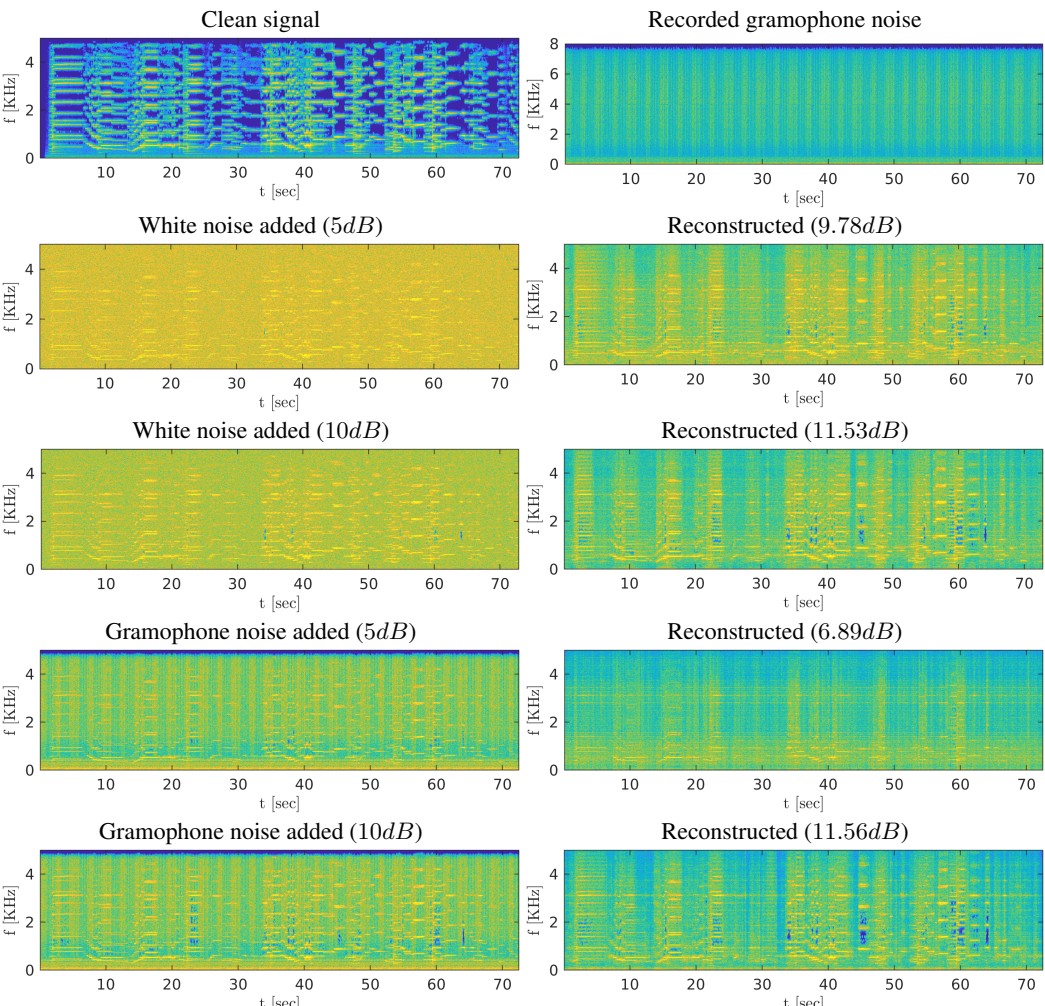

Figure 8: **Audio denoising.** The upper row depicts the original violin recording (left) and the recorded gramophone noise (right). The other rows show results of denoising white and recorded noise, at two levels of input SNR.

Instructions to participants

**About this HIT:**

- **Please only participate in this HIT if your computer has output audio device (headphones/speakers).**

- It should take about 10 minutes.

- You will take part in an experiment involving auditory perception. You'll hear a series of audio signals. Each of them is either a real signal or a "fake" signal that was generated using a computer program. Your task is to determine weather the signal is real or fake. Sometimes the fake signal may sound very plausible.

- You will complete a short practice (less than 1 minute) before starting the main task.

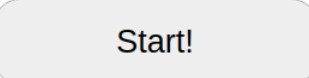

*By making judgments about these sounds, you are participating in a study being performed by scientists. Your participation in this research is voluntary. You may decline further participation, at any time, without adverse consequences. Your anonymity is assured; the researchers who have requested your participation will not receive any personal information about you.*

Question presented to participants

Do you think this sound was **fake**?
(answer "Yes" if you think it was fake, or "No" if you think it was a real one)

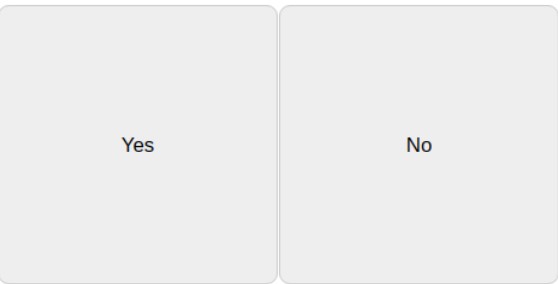

*Practice trial 1 out of 5*

Figure 9: **Unconditional generation unpaired user study.** After reading instructions (upper) and listening to a sound sample one time, the participant had to answer whether this sound was fake (bottom).

## Instructions to participants

**About this HIT:**

- **Please only participate in this HIT if your computer has output audio device (headphone/speaker).**

- It should take about 5 minutes.

- You will take part in an experiment involving auditory perception. You'll hear a series of pairs of audio signal. In each pair, one signal is a real while the other signal is "fake", and was generated using a computer program. Your task is to determine which signal is fake. Sometimes the fake signal may sound very plausible.

- You will complete a short practice (less than 1 minute) before starting the main task.



Start!



*By making judgments about these audio samples, you are participating in a study being performed by scientists. Your participation in this research is voluntary. You may decline further participation, at any time, without adverse consequences. Your anonymity is assured; the researchers who have requested your participation will not receive any personal information about you.*

## Question presented to participants

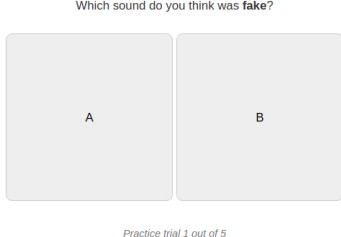

Which sound do you think was **fake**?

*Practice trial 1 out of 5*

## Feedback after tutorial questions

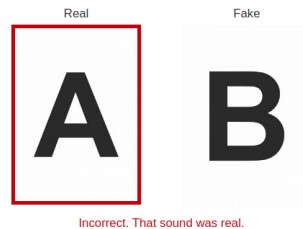

Incorrect. That sound was real.

Figure 10: **Unconditional generation paired user study.** After reading instructions (upper) and listening to real and fake sounds one time each, the participant had to decide which sound was fake (middle). The bottom image shows example of the paired tutorial presented to participants.

<p style="text-align:center">Instructions to participants</p>

**About this HIT:**

- **Please only participate in this HIT if your computer has output audio device (headphone/speaker).**

- It should take about 10 minutes.

- You will take part in an experiment involving auditory perception. You'll hear a series of audio signals with a gap of silent within them, meaning a part of the signal is missing. Right after, you will hear two possible completions for the gap. Your task is to determine which of the two options better complete the gap. You will be able to re-listen to the signals as much as you'd like. Sometimes both options may sound very plausible.

<p style="text-align:center">Start!</p>

<p style="text-align:center">Question presented to participants</p>

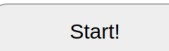

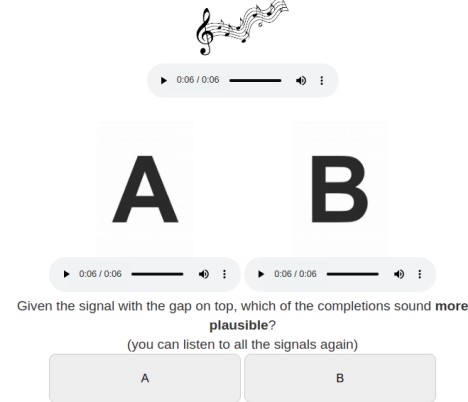

Given the signal with the gap on top, which of the completions sound **more plausible**?
(you can listen to all the signals again)

| A | B |
|---|---|

Figure 11: **Inpainting user study.** After reading instructions (upper), participants were given the sound with a gap, along with two possible completions. They could listen to all three signals as many times as they wanted, and had to decide which completion sounded better.