# OpenReview forum: "Catch-A-Waveform: Learning to Generate Audio from a Single Short Example"
_NeurIPS.cc/2021/Conference — NeurIPS 2021 Poster_

### Official Review · Reviewer_r27W · 2021-07-01

**Rating:** 5
**Confidence:** 4

**Summary:**

They main goal of this article is to investigate if it's possible to build a (specific) generative audio model out of short audio snippet.
They use a "progressive GAN"-ish model with an additional reconstruction loss, to explore this paradigm. Thei experiment with a fair amount of use cases, demonstrating some degree of success.

**Limitations And Societal Impact:**

Limitations: Its main limitation is its synthesis quality – the model introduces noticeable artefacts.

Societal impact: If further developed, this model could allow "personalised deep learning models" – what can allow for private models that could be trained on the client side with just few data. That said, the computational requirements might be challinging – since "training on a 25 second long signal takes about 10 hours on Nvidia GeForce RTX 2080".

**Main Review:**

The paper shows that it is possible to build a (specific) generative audio model out of short audio snippet. However, the artefacts I can hear (in their great webpage with examples!) are the ones I personally associate with the first generation of waveform-based models – that were trained with little data and with small capacity models. Hence, while the model seems to work (up to some extend) it synthesises audio with a fair amount of artefacts. In line with that, one comment:
- "For the task of bandwidth extension, only a single 20 second long full-bandwidth training recording typically suffices for obtaining high quality results." -> Under my view, those are not high quality speech renditions – please rephrase. That said, the article "Bandwith extension is all you need" might be of relevance for the authors.

Regarding the "structure" of the new renditions:
- "new signals that are semantically similar to the training recording, but contain new compositions and structures" -> Under my view, the resulting music lacks of "structure" – in the same way the speech signals are "not necessarily interpretable" becuase the "model has no notion of language". For this reason, I think this model does not shine for music or speech signals – but the ambient sounds are super convincing.
- "generate new realistic improvisations" -> Similarly, I don't think the improvisations are realistic because they lack structure and have artefacts. Please, rephrase.
- I also miss a discussion on why several examples "repeat" a variation of the training audio at the beginning of the rendition.

Comments related to the perceptual tests:
- "are often confused to be real" -> That's interesting, because your perceptual tests (more-or-less) support this idea, but when I hear to the audio samples myself I think the synthesised audio has artefacts (what makes them clearly distinguishable from the originals).
- Have you considered running a statistical test to see if there is a statistical difference against (50%, 50%) no preference?
- "In fact, our evaluation suggests that limiting the training to a single short signal is actually beneficial, and can outperform models trained on hours of recordings." -> I'm not sure how you back this claim.

Related to the model:
- "we take the variance of zn to be proportional to the energy of x in the frequency band" -> Have you considered injecting less noise, also? The audios sound very noisy. What if you just sample the zn noise at the coarsest level? Or to sample with a dropout layer?
- I enjoyed the conditional generation and bandwith extension experiments, is a smart way to use the model at hand to build "personalized deep learning models".
- Have you considered pre-training the model with more data? And then finetuning for the specific use case?

Minor comments:
- "This is despite its complete lack of prior knowledge about the nature of audio signals in general." -> I'm not sure I understand what you mean here.
- I enjoyed the failure cases! Thanks for sharing – it was very useful.

**Time Spent Reviewing:**

3

---

> ### Author Response · Authors · 2021-08-10
> **Thanks for the thorough review and detailed comments**
>
> **High quality in bandwidth extension:**
>
> What we meant here is that performing BE using our method, with only 20 seconds of training audio, provides results that are comparable and even superior to the state of the art in this task (namely, instead of “high quality”, we mean “higher quality than existing methods''). We’ll rephrase. Please note that the paper you mention, as well as many others (see e.g. [54,19] in the main text), deal with a much easier BE problem, where the input signal is sampled at a relatively high rate to begin with (usually 8Khz). In our evaluation, we targeted the more challenging task of BE for inputs sampled at 4Khz. The difference between these settings is very significant when it comes to speech signals, as the unvoiced consonants are completely lacking at 4Khz. Following your comment, we’ll add illustrations of BE for signals sampled at 8KHz, where the quality that we get is naturally higher.
>
> **Generated music lacks structure:**
>
> We completely agree that our generated music signals are not structured in a way that a human composer would achieve. We will add a discussion about this. Nevertheless, we are able to see that thanks to learning at multiple receptive fields (via our multi-scale architecture), our model generates sequences of notes that aren’t present in the original recording, while avoiding significant discontinuities. As suggested by Reviewer Bdte, we will add a similarity matrix visualization between generated and real signals. This will show that on short timescales there exist similar segments, whereas on longer timescales, the generated signals contain new compositions. Again, we acknowledge that these new structures are not guaranteed to be harmonically pleasing when it comes to music generation, and we will add a discussion of this.
>
> **Realistic improvisations:**
> We’ll rephrase. Nevertheless, please note that the task of composing long coherent musical compositions is non-trivial even for models trained on hours of recordings, and is still being extensively investigated. We claim that succeeding in creating ‘fairly’ coherent structure with such little training data, is still very relevant for the audio-processing community.
>
> **Generated signals repeat a part from the training audio at the beginning:**
>
> This is a great point. This phenomenon is related to the padding of the input noise. We use zero padding to ensure the generator’s output is of the same length as the real signal during training. This zero padding functions as a positional encoding, which allows the generator to know the absolute location with respect to the beginning and ending of the signal. This encoding, however, extends only up to one receptive field from the beginning and one receptive field from the ending of the signal. Therefore, the generator manages to “remember” these parts and “paste” them at the borders of the generated signals. If the user prefers, this can be avoided by simply trimming the generated signal by one receptive field from each side. Note that this is a known phenomenon (see the supplementary material of [50] in the main text, as well as [R1] below). We will explain this in more detail in the camera ready version.
>
> [R1] Xu, et al. "Positional encoding as spatial inductive bias in GANs," CVPR’21.
>
> **Statistical difference against 50% preference:**
>
> A Binomial test rejects the hypothesis of 50% preference with high confidence in all our experiments. Thus, the confusion rates are high, but still lower (in a statistically significant way) than 50%. We’ll also add [5%,95%] confidence intervals according to the Wilson method. For the paired test, we have [35.4%, 40.8%], for the unpaired test w/ paired tutorial we have [42.0%, 47.6%] and for the unpaired test w/ unpaired tutorial [44.5%, 50.1%].
> We believe that the high confusion rates are due to the general population of listeners that participated in the user studies. It is likely that had we performed studies with musicians / sound-technicians / audio-researchers, we would have seen lower confusion rates. Unfortunately, collecting a large enough pool of such listeners is non-trivial.
>
> **Limiting the training to a short signal is beneficial:**
>
> This claim refers to inpaining and bandwidth extension, where our model outperforms models that were trained on hours of data. We agree that at the point this is mentioned in the paper, it is a bit vague. We’ll add concrete references to the later sections of the paper that illustrate this.
>
> **Try injecting less noise:**
>
> This is actually an interesting point. We indeed tried to inject less noise and found that it wasn’t helpful in reducing the noise in the generated signals. The model sometimes generates noisy outputs, regardless of the amount of noise injected in the input. We believe this issue can be improved in future work by investigating more complex architectures. Another point worth mentioning is the tradeoff between the amount of noise injected and the variability of the generated signals. This point is briefly discussed in Section 4.2 in the paper.
>
> **Training with more data and fine-tuning on a specific signal:**
>
> Our goal in this paper was to explore the limits of what can be done with a single short audio track. We completely agree that once this is fully understood, it is natural to combine our technique with some sort of external training on a large dataset. We leave this for future work.
>
> **Complete lack of prior knowledge about audio:**
>
> The intent here was that our model has not been exposed to any other audio signal besides the short training signal. Thus, it doesn’t know how speech sounds like in general, or how music sounds like in general. It is only trained on a few seconds of a single signal. This is in contrast to models trained on hours of speech recordings or music, which can learn notions of language or rules of harmony.
>
> **Societal impact:**
>
> The idea of personalized deep learning is indeed interesting. We’ll mention it. We might see future work achieving improved results on even smaller datasets. Furthermore, the computational issues may be improved in the future by investigating more lightweight architectures.

---

### Official Review · Reviewer_Bdte · 2021-07-16

**Rating:** 9
**Confidence:** 5

**Summary:**

This paper presents a strategy for learning to generate convincing audio using only a short audio example as training data. The method involves iteratively upsampling the audio and shaping whitre noise, a procedure which lends itself to variable-length unconditional synthesis as well as several downstream conditional generation tasks such as bandwidth extension and inpainting.

**Limitations And Societal Impact:**

The authors explicitly address some limitations of their method in the main text of the paper. The societal implications are not discussed however. Some discussion of this would be nice to see for the camera ready - I think there are a lot of applications of this work that could benefit society.

**Main Review:**

Overall, this is an excellent paper and the results are quite impressive. I have no doubt that this will be an influential paper within the field of deep audio synthesis. The writing is also very clear and easy to follow, and the method is simple and intuitive. Additionally, the paper appears to have put forth (but not advertised) a solution to a major issue with past work on audio synthesis: the ability to generate audio of arbitrary length with continuity across receptive field chunks.

Ironically, while the paper primarily advertises its strengths at unconditional generation, I am generally more convinced by the _conditional_ generation tasks like music variation generation, bandwidth extension, inpainting, etc. The unconditional generation sounds much more like recombination than generalization. This isn’t necessarily a major criticism as recombination of audio with local continuity is difficult to achieve, as small discontinuities which occur from naive recombination can be perceptually devastating. However, it does sound like the model is just stitching together short windows of audio, not unlike what can be accomplished with traditional granular audio synthesis (in ML parlance, granular synthesis might be called a “retrieval-based” method).

One analysis that would be very useful to see is a similarity matrix between the ground truth and the generated waveform in some audio feature space. I suspect such a figure would make it much easier (compared to listening) to determine the temporal correspondence between the generated result and the original training excerpt, which would either help confirm or discredit my hypothesis that the model is stitching together copied segments of the training data. However, even if it is the case that it is mostly copying, I reiterate that this is still nontrivial, and I would prefer to see the limitation addressed rather than swept under the rug.

Regardless of potential limitations in the unconditional setting, the number of conditional generation tasks that this model excels at is quite impressive. The proposed method appears ready for immediate application, and will be of interest to audio engineers, musicians, and sound artists alike.


**Time Spent Reviewing:**

2

---

> ### Author Response · Authors · 2021-08-10
> **Thanks for the great feedback**
>
> **Add similarity matrix visualization:**
>
> This is a good idea. We’ll add it. We completely agree that on small time-scales, the auditory elements generated by our model have high similarity to those in the training signal. Indeed, it is extremely difficult to generalize to new structures from so little training data. On longer time scales, the model does compose new structures from these basic short building blocks. But we acknowledge that these new structures are not necessarily semantically meaningful, because the model can’t learn the rules of English or the rules of harmony from just a few seconds of training audio. We will add a more detailed discussion of this.
>
> **Societal implications:**
>
> As suggested by Reviewer r27W, our method enables a sort of ‘personalized deep learning’, where users can manipulate their own small data in generative tasks. We believe this use case can be implemented in many applications, which can have both positive and perhaps negative societal impacts. We will discuss these issues in more detail in the camera ready version.

---

### Official Review · Reviewer_Qtuh · 2021-07-22

**Rating:** 8
**Confidence:** 5

**Summary:**

The authors demonstrate that a SinGAN model can be adapted for audio, training a model on a single instance of audio, and generating continuations and inpainting in both the time and frequency domain. They utilize a multi-scale decomposition to create a pyramid of generators and discriminators, and also force reconstructions for one of the points of the generator's latent space. They make several model accomodations to training on the audio domain, including a criteria for automatically tuning the selected scales of their pyramid to the low-frequency power content of the target audio. They show many applications for their technique, including unconditional temporal continuations, conditional regeneration of higher frequency content to create new song sections, bandwidth extension of low resolution audio, inpainting of audio segments, and implicit denoising through the reconstruction signal. Qualitative and quantitative results support their findings, including listening tests and an extensive page of examples.

**Limitations And Societal Impact:**

++ It is great that limitations of the quality of the generation are considered in both the main text and the supplement, and the examples provided are very informative.

-- That said, there seem to be stronger limitations to the technique in the audio domain that are less perceptually relevant in the image domain. In particular, certain high-level structure such as language (both in english and in terms of musical notes) are not captured by training on a single example. For example the song continuations created garbled pseudo-language mixes of phonemes because they haven't seen words not present in the original clip. Similar for the musical note motifs of continuations, they are more "smoothed concatenations" of the original segments than a true improvisation would consider (there's only so much high-level structure that can be present in a small clip of audio). This is a fundamental limitation of this technique that is less relevant for SinGAN on images as the application of continuing the texture of a mountain skyline of fleet of hot air balloons doesn't require generating novel high-level structure (say a building on the hillside or airplanes flying among the balloons). That said, generating new song segments or improvisations does seem to require new high-level structure (new words, new motifs etc.) and this technique will be limited in these applications unless it is modified to incorporate this outside knowledge from elsewhere. Further highlighting and exploration of these limitations would give a better sense for what this technique does and does not do.

+/- There are not any discussions of Societal Impact, which would be a problem if the model were more convincing at speech continuation, as voice cloning is worth considering as a negative impact. That said, the limitations mentioned above about high-level structure and control limit the possible impact of this particular work.

**Main Review:**

Originality / Significance


-- Straightforward extension of previous SinGAN work to a new domain, so novelty of the approach is limited.

++ However, the authors do a good job of adapting the technique to the audio domain, and demonstrating that it is possible, which is not a forgone conclusion. Further, the breadth of creative applications they demonstrate are of broad interest to the community and show the further potential of the original SinGAN architecture.

++ In particular, the authors provide several clever adaptations of the model (such as generating new song sections by conditionally sampling higher frequencies, and denoising by using the reconstruction signal) that provide new insight into the domain and are not at all straightforward to expect from this model.


Quality / Clarity

++ The paper is clearly written, laid out in a logical manner, with ample figures and examples to demonstrate the many examples and applications proposed.

++ Quantitative results are presented, including listening tests for some of the more qualitative examples such as the quality of continuations.

-- While it is helpful to have the paired / unpaired / tutorial splits in Table 1, the study is lacking a reasonable baseline comparison, such as extending the audio by concatenating (with crossfades) audio chunks from the original audio. A pairwise comparison, or even mean opinion score comparison, would be informative for the quality of the technique rather than the current Turing test of "Is this audio fake?"

-- More quantitative metrics for the audio denoising experiments would help to understand the quality of the denoising. There are many standard denoising benchmarks where noise is added to clean signals and SNR can be computed.



**Time Spent Reviewing:**

4

---

> ### Author Response · Authors · 2021-08-10
> **Thanks for the constructive comments**
>
> **User studies are lacking a reasonable baseline:**
>
> Thanks for the suggestion, we’ll add a naive baseline of the sort you suggest. Just note that our paired user studies are not Turing tests of the type “Is this audio fake?”. They are in fact of the form you suggest, only with a tougher baseline - the ground-truth (original) audio. Namely, users listened to both the real signal and the fake audio generated from that signal, and had to determine which is which. Since the confusion rate in this test was quite high (nearly 40%), we believe that comparing to any naive baseline, will result in preference to our method. In any case, we’ll add this test.
>
> **Quantitative metrics for the audio denoising experiments:**
>
> It’s worth pointing out that the point of this section was to demonstrate the ability of enhancing an old recording where a clean version does not exist. In such settings the “noise” is not necessarily an additive perturbation. Therefore, experiments on datasets with additive noise can only tell part of the story about CAW’s ability to enhance old recordings. However, we’ll add experiments where we have access to a clean ground-truth recording and SNR can be calculated.
>
> **Highlighting limitations relating to lack of high-level structure:**
>
> We completely agree, and will try to better highlight this point in the paper. Naturally, generating new high-level information is limited in our method, as the model has access to very little training data. We had no intent of hiding this limitation. We thought (perhaps mistakenly) that this was clear from the problem setting, as it’s impossible to learn English from only a few sentences, or to learn rules of harmony from only 20 seconds of music. This is why we rather chose to focus on the surprising things that still can be done in this challenging setting. Following your comment, we’ll certainly try to balance our exposition and better highlight the fundamental limitations of learning from a short audio signal.

---

> > ### Comment · Reviewer_Qtuh · 2021-08-31
> > **Thank you for your response**
> >
> > Thank you for the thoughtful response. I think adding these changes will help make the paper stronger. They have been helpful for reviewer discussion, and my score remains the same. I feel the paper is a strong contribution to the literature, but that the prose is a bit over aggrandizing given the results, which are still strong. Focusing the paper on what it does and does not do well, with more measured statements, should better show off the valuable work.

---

### Decision · Program_Chairs · 2021-09-27

**Decision:**

Accept (Poster)

**Comment:**

The paper introduces a way to build a specific generative audio model out of short audio snippet, via a GAN-based approach. While the proposed approach is combining existing techniques in the field (as it's mostly derived from SinGAN), reviewers agree that the adaptation to speech is novel enough.

A shortcoming of the approach is the artifacts generated by the model - however, a majority of the reviewers found that the quality is good enough.

The authors should include the changes promised in the rebuttal - in particular, it was mentioned during the discussion period that overselling points on the efficacy and quality of the approach should be toned down.